# Implications of Noise on Neural Correlates of Consciousness: A Computational Analysis of Stochastic Systems of Mutually Connected Processes

**DOI:** 10.3390/e23050583

**Published:** 2021-05-08

**Authors:** Pavel Kraikivski

**Affiliations:** Academy of Integrated Science, Division of Systems Biology, Virginia Polytechnic Institute and State University, Blacksburg, VA 24061, USA; pavelkr@vt.edu

**Keywords:** neural correlates of consciousness, spectral entropy, power spectrum, stochastic modeling, noise in neuronal networks

## Abstract

Random fluctuations in neuronal processes may contribute to variability in perception and increase the information capacity of neuronal networks. Various sources of random processes have been characterized in the nervous system on different levels. However, in the context of neural correlates of consciousness, the robustness of mechanisms of conscious perception against inherent noise in neural dynamical systems is poorly understood. In this paper, a stochastic model is developed to study the implications of noise on dynamical systems that mimic neural correlates of consciousness. We computed power spectral densities and spectral entropy values for dynamical systems that contain a number of mutually connected processes. Interestingly, we found that spectral entropy decreases linearly as the number of processes within the system doubles. Further, power spectral density frequencies shift to higher values as system size increases, revealing an increasing impact of negative feedback loops and regulations on the dynamics of larger systems. Overall, our stochastic modeling and analysis results reveal that large dynamical systems of mutually connected and negatively regulated processes are more robust against inherent noise than small systems.

## 1. Introduction

Noise is ubiquitous in neuronal circuits [1,2]. Hence, neuronal networks may produce highly variable responses to the repeated presentation of the same stimulus [3,4]. The dominant sources of noise in neuronal systems include voltage-gated channel noise [5,6] and synaptic noise [7,8], as well as sensory-motor sources of noise [2]. Several studies have examined the implications of noise on membrane potential [1], propagation of action potential [9], and spike train coding [10].

In bistable and threshold-like systems, noise can significantly affect the information-processing of sub-threshold periodic signals that are more likely to cross the threshold in the presence of noise. Therefore, the propagation of weak periodic signals can be enhanced by the presence of a certain level of noise. This phenomenon is called stochastic resonance. The noise-induced transmission of information has been observed in cat visual neurons [11], rat [12], and crayfish [13]. Several theoretical studies of information-processing in threshold-like neuronal systems show that additive noise can increase the mutual information of threshold neurons [14,15,16]. Stochastic resonance can also modulate behavioral responses. For example, noise has been shown to improve human motor performance and balance control [17] and enhance the normal feeding behavior of river paddlefish [18]. Therefore, noise can influence perception and behavior by altering the information transmission and processing in nervous systems.

Overall, noise is a key component of the sensorimotor loop and, thus, may have direct behavioral consequences [2]. Noise may also increase the information capacity of neuronal networks and maximize mutual information [16]. Average mutual information across all bipartitions of a neural network is used as a metric for neural complexity in the information integration theory of consciousness [19,20]. Thus, noise can be directly linked with conscious perception. Undoubtedly, it is also important to consider noise in the neural correlates of consciousness studies that pave the way for understanding neural mechanisms of perception and consciousness [21,22,23,24]. Neural correlates of consciousness are identified by the minimal set of neuronal events sufficient for any one specific conscious percept [23]. Noise can have a substantial impact on the system dynamics that must be considered when the minimal set of neuronal events sufficient for any one specific conscious percept is defined.

In this work, we study the implications of stochastic noise on systems of processes exhibiting dynamics isomorphic with a specific conscious percept [25,26]. During the past few decades, several comprehensive theoretical models of consciousness have been developed based on dynamical and operational system frameworks [27,28,29,30], information theory [19,31], temporo-spatial theory [32,33], and several other mathematical and physical frameworks [34]. It is widely accepted that consciousness is a dynamic process that requires the proper execution of physical neuronal operations [19]. Conscious states are observed only when both the neuronal processes are correctly executed and running and the connections between different parts of neuronal networks (brain areas) are largely maintained and functional [35]. It has also been proposed that the specific conscious percept can be associated with a particular functional organization of a neuronal system [36], or that the operations of transient neuronal assemblies must be functionally isomorphic with phenomenal qualities [28]. Our dynamic model exhibits this important property.

To study the implication of noise on neural correlates of consciousness, we developed a stochastic model based on our previous dynamical system framework for a conscious percept [25,26]. The main purpose of this work is to investigate how noise affects the dynamical systems of different sizes. We computed the spectral entropy for different systems and determined the dependence of entropy on system size. Overall, we believe that our stochastic model can help us better understand the impact of noise on neural correlates of consciousness. Understanding the impact of noise on perception can help us deduce processes related to conscious perception, apply this knowledge to modulate behavioral responses [17,18] and design computer vision systems [37,38].

## 2. Materials and Methods

To generate stochastic trajectories for processes, we used Gillespie’s stochastic simulation algorithm. This method is often applied to handle stochastic fluctuations in chemical reaction systems [39,40,41]. In Gillespie’s formalism, the kinetic terms describing rates of biochemical reactions are treated as stochastic propensities for a reaction event to occur over a small interval of time. This algorithm is the most accurate for elementary reaction mechanisms expressed in terms of mass-action rate laws [39]. However, the algorithm has been also used to handle stochastic fluctuations in chemical reaction systems described by other phenomenological rate laws (like Hill function) that are commonly used in deterministic models of biochemical systems [40]. Overall, Gillespie’s scheme is often considered as a convenient way to turn a deterministic simulation into a stochastic realization by reinterpreting kinetic rate laws as propensities of a reaction. Gillespie’s stochastic simulation algorithm has been also applied to simulate reaction-diffusion systems [42], the predator-prey cycles [43], and the evolutionary dynamics of a multi-species population [44]. Here we use Gillespie’s scheme to introduce randomness into the system of processes which we had previously described by deterministic differential equations [25].

The deterministic systems (2, 3) were converted into stochastic models using the Gillespie method, where the right-hand-side terms of ordinary differential equations describe propensity functions determining the probability for a process $p_{i}$ to increase, decrease, or transition to a process $z_{i}$. The algorithm can be described by the following general Gillespie scheme [39]:Initialize the process state vector, $\vec{P}$, and set the initial time at 0.Calculate the propensities, $a_{k}\left(\vec{P}\right)$.Generate a uniform random number, $r_{1}$.Compute the time for the next event, $\tau =-\frac{1}{\sum_{k}a_{k}\left(\vec{P}\right)}\ln r_{1}$.Generate a uniform random number, $r_{2}$.Find which event is next, I=i, if $\frac{\sum_{k=1}^{i-1}a_{k}\left(\vec{P}\right)}{\sum_{k}a_{k}\left(\vec{P}\right)}\leq r_{2}<\frac{\sum_{k=1}^{i}a_{k}\left(\vec{P}\right)}{\sum_{k}a_{k}\left(\vec{P}\right)}$Update state vector, $\vec{P}\to \vec{P}+y_{i}$.Update time, t→t+τ.Repeat (2)–(8).

XPP/XPPAUT software (http://www.math.pitt.edu/~bard/xpp/xpp.html, accessed on 24 March 2021) was used to solve systems of ordinary differential equations, compute one-parameter bifurcation diagrams (Figure 3A), and simulate stochastic models. XPPAUT codes that were used to simulate results in Figures 1, 3A and 4A,D,G are provided in Appendix A.

The eigenvalues shown in Figure 3B were computed using Wolfram Mathematica software.

To implement Gillespie’s stochastic simulation algorithm in the XPPAUT codes (provided in Appendix A), we first computed sums of event propensities $x_{i}=\sum_{k=1}^{i}a_{k}\left(\vec{P}\right)$ for all values of i, then we found the next event (steps 5, 6), updated the time (4, 8) and the states of processes $\vec{P}$ (step 7).

We used the spectrum analysis and spectral entropy to quantify noise effects in the system of processes. These techniques are common tools that are often applied to analyzed data obtained in neurophysiological studies [45,46,47,48]. These tools are also commonly applied in signal processing, control systems engineering and diagnosis. For example, spectrum techniques are often used to monitor dynamics of complex machines and their fault diagnosis [49,50]. Spectral entropy in our work is based on Shannon’s entropy formalism that is a foundational concept of information theory [51]. The entropy metric is also an essential part of information integration theory of consciousness [19,31]. Particularly, the entropy metric is used to quantify integrated information of a system composed of interacting elements. We used the spectral entropy to quantify the effects of noise on the system of interacting processes and how the impact of noise changes when the system size increases.

The fast Fourier transform (FFT) of each process p(t) was computed using the Fourier Analysis function in Excel’s Analysis ToolPak. 1024 points were used to obtain the signal p(f) in the frequency domain. 1024 points corresponded to a total simulation time of ~370 arb. u. and the period of oscillations ranged between ~5–10 arb. u. (see Figure 4C,F,I). The sampling frequency, f, was computed by dividing the number of points by the time interval, Δt. The frequency magnitudes were calculated using Excel’s IMABS function. The power spectral density was defined as $PSD_{i}={\left|p\left(f_{i}\right)\right|}^{2}/2\Delta f$. We used 512 data points to compute spectral densities. The spectral entropy was computed using the following equation:(1)$$S=-k\sum_{i=1}^{512}\hat{PSD_{i}}{\mathrm{Log}}_{2}\left({\hat{PSD}}_{i}\right),\;$$
where $k=\frac{1}{{\mathrm{Log}}_{2}\left(512\right)}\approx 0.1$ and $\hat{PSD}$ is the normalized power spectral density. $\hat{PSD}$ was computed by dividing the power spectral density by the total power [52].

## 3. Results

We considered a dynamical model that describes mutually connected processes. In this model, the processes maintain specific dynamic relationships, which are designed to be isomorphic with a hypothetical conscious percept to mimic a mechanism for neural correlates of consciousness [26]. The specific conscious percept was represented as a specific function (property) that was performed and maintained by the dynamical system. This is in line with William James’s interpretation of consciousness as a dynamical process, not a capacity, memory, or information [53].

A nonlinear system of two processes and linear systems of four, eight, and sixteen processes were used to mimic some specific conscious percepts exhibited by these dynamical systems. The former system was used to illustrate the concept that the system can execute and maintain a specific relationship between processes $p_{1}=f\left(p_{2}\right)$ and $p_{2}=f\left(p_{1}\right)$ where f(…) is a nonlinear function. The latter system was used to investigate the effect of the size of the system. In the linear system of mutually connected processes, each process could be expressed through all other processes as $\vec{P}=A\vec{P,}$ where $\vec{P}$ is a vector of processes and A is the hollow matrix. The corresponding stochastic models were used to investigate the effects of noise on these systems.

First, we consider a system of two nonlinear differential equations:(2)$$\frac{dp_{1}}{dt}=-p_{2}+p_{1}\left(R^{2}-p_{1}^{2}-p_{2}^{2}\right)\;\frac{dp_{2}}{dt}=p_{1}+p_{2}\left(R^{2}-p_{1}^{2}-p_{2}^{2}\right).$$
System (2) has a periodic solution $p_{1}=R\;\cos\left(t\right)$, $p_{2}=R\;\sin\left(t\right)$ shown in Figure 1A. The solution could also be represented by a limit cycle in the $p_{1}p_{2}$-phase plane (see Figure 1B). Thus, the relationship between the two processes is defined by the limit cycle, which is maintained in time. The parameter *R* defines the radius of the limit cycle. For System (2), the specific dynamic relationship between $p_{1}$ and $p_{2}$ is isomorphic with a circular motion. This relationship itself is a part of the dynamical system. We assume that the specific conscious percept is represented by the dynamical property that emerges in the dynamics of the system, which, in this case, is isomorphic with a circular motion.

We introduced noise into System (2) by applying a stochastic formulation. Here, we used the Gillespie algorithm, described in the Materials and Methods section, which allowed us to describe the evolution of processes using propensities derived from rates that govern the dynamics of the processes in System (2). The simulated stochastic trajectories for the *p*_1_ and *p*_2_ processes are shown in Figure 1C and the corresponding limit cycle in the *p*_1_*p*_2_-phase plane is shown in Figure 1D. Using stochastic simulation results, we computed power spectral densities (see Figure 1E) from stochastic trajectories as well as the spectral entropy using Equation (1) as described in the Materials and Methods section. For the stochastic version of a nonlinear system (2), we found that spectral entropy was ~0.5 for both the *p*_1_ and *p*_2_ processes.

Nonlinear relationships among the processes are expected for any nonlinear system and a system consisting of more than two processes could represent a challenge for mathematical analysis. Thus, to investigate how system size alters the impact of stochastic noise on a system, we used a scalable linear system of interacting oscillating processes and we analyzed a system of coupled oscillators that was described by a set of linear differential equations [25]. This system of coupled oscillators represents a set of interacting processes that have the following two properties: (i) each process in the set could be represented by a linear combination containing all other processes in the set, and (ii) the relationships among the processes are isomorphic to a distance matrix. We then developed a stochastic model describing this system to study the implications of noise on a system of mutually connected processes. We consider two sets of n-processes, $\vec{P}=\left(p_{1},p_{2},\ldots ,p_{n}\right)$ and $\vec{Z}=\left(z_{1},z_{2},\ldots ,z_{n}\right)$, which are described by the following system of equations:(3)$$\frac{d\vec{P}}{dt}=A\vec{P}-\left(\vec{Z}+\vec{P}\right)\;\frac{d\vec{Z}}{dt}=\vec{P}.$$
The deterministic system (3) was extensively analyzed in Ref. [25]; in this paper, our goal was to analyze the stochastic version of System (3). System (3) has oscillatory solutions such that $\vec{P}=A\vec{P}$, where ***A*** is a hollow distance matrix [25]. Matrix ***A*** defines how each process $p_{i}$ in the system is connected to all other processes, and we considered the following relationship between processes:(4)$$p_{i}=\sum_{j=1}^{n}{\left(i-j\right)}^{2}\varepsilon p_{j}$$
and thus
(5)$$A\;=\;\left(\begin{array}{ccc} \begin{array}{c} 0\\ \varepsilon \end{array} & \begin{array}{c} \varepsilon \;\;\;\;\ldots \\ 0\;\;\;\;\;\;\ldots \end{array} & \begin{array}{c} {\left(n-1\right)}^{2}\varepsilon \\ {\left(n-2\right)}^{2}\varepsilon \end{array}\\ \vdots & \;\;\;\vdots \;\;\;\ddots & \vdots \\ {\left(n-1\right)}^{2}\varepsilon & {\left(n-2\right)}^{2}\varepsilon \cdots & 0 \end{array}\right).$$

Figure 2 shows a wiring diagram presenting the relationships among processes described by System (3).

System (3) has one parameter *ε* that is considered as a bifurcation parameter for the system (3). The typical one parameter bifurcation diagram for system (3) is shown in Figure 3A. The Hopf bifurcation value of *ε* depends on the number of processes in the system. For systems consisting of 4, 6, …, 20 processes, the Hopf bifurcations occurs at *ε* equals to ±1, ±1/4, ±1/10, ±1/20, ±1/35, ±1/56, ±1/84, ±1/120, and ±1/165, respectively. For a system of two $p_{i}$ and two $z_{i}$ processes, the stability of the steady states of system (3) is described by four eigenvalues. The real and imaginary parts of these eigenvalues, as a function of parameter *ε*, are shown in Figure 3B. The real parts of all eigenvalues are negative for −1<ε<1, indicating a spiral sink for this range of *ε*. For |ε|>1 the spiral source is observed. For ε=±1 the system exhibits oscillations with a constant amplitude.

Noise was introduced into system (3) by applying the Gillespie stochastic formulation described in the Materials and Methods section. The stochastic model was then used to investigate how noise affects the dynamical systems that consists of a different number of processes. We performed simulations for a system of four ($p_{1}$, $p_{2}$, $z_{1}$, $z_{2}$), eight ($p_{1}$,…, $p_{4}$, $z_{1}$,…, $z_{4}$), and sixteen ($p_{1}$,…, $p_{8}$, $z_{1}$,…, $z_{8}$) processes that interact as shown in Figure 2. Stochastic trajectories for processes $p_{i}$ in three systems consisting of two, four, and eight $p_{i}$ processes are shown in Figure 4A,D,G, respectively. Note that the number of $z_{i}$ processes in the system is always the same as the number of $p_{i}$ processes. However, we only describe the dynamics of the $p_{i}$ processes because their dynamics exhibit the property that each process $p_{i}$ has a specific relationship with all other processes (see Equation (4)), and this relationship is isomorphic with the distance matrix. Figure 4 also shows the distribution functions (see Figure 4B,E,H) and the normalized power spectral densities (see Figure 4C,F,I) computed using stochastic trajectories for the selected process $p_{2}$. The method to compute and normalize power spectral densities is described in the Materials and Methods section. For other processes included in the system, the distribution functions and power spectral density plots look nearly identical to those shown for the process $p_{2}\;$ in Figure 4.

To quantify the implications of noise on systems of different sizes, we computed spectral entropy values for all processes in the systems using Equation (1). The results are summarized in Table 1. We observed that the spectral entropy values decrease as the number of processes in the system increases. Figure 5 shows the average spectral entropy values as a function of system size, which is represented by the number of processes $p_{i}$ constituting the system. Interestingly, spectral entropy decreases linearly as the system size doubled. This indicates that a larger system is more robust against the influence of inherent fluctuations than smaller systems.

## 4. Discussion

In this paper, we studied how inherent noise impacted the dynamical systems that mimic a mechanism for neural correlates of consciousness. Our modeled system exhibits a dynamic behavior that is isomorphic with a specific conscious percept [25,26]. Details on how phenomenal conscious states are assumed to arise in dynamical systems are provided in our previous works [25,26]. The neural correlates of consciousness are defined as a minimal mechanism sufficient for any one specific conscious percept. Here, our analysis is concentrated on implications of noise on the mechanisms that are scaled to different sizes. To study and characterize noise effects on the dynamic behavior of the mechanism, we developed a stochastic version of our deterministic model described in Ref. [25].

The main finding of our study is that the larger system of mutually connected and negatively regulated processes is more robust against the influence of inherent fluctuations. We found that spectral entropy of the system decreases linearly as system size doubled (see Figure 5 and Table 1). In addition, we found that the frequency domain, for which the power spectral density values are significant, shrinks as system size increases (see Figure 4C,F,I). This indicates that the noise impact is more restricted when the number of regulatory feedback loops (see Figure 2) in the mechanism increases. Our results agree with several other experimental and computational studies of noise in biological regulatory circuits, which revealed that negative feedback loops suppress the effects of noise on the dynamic behavior of the circuits [54,55].

Comparing the power spectral densities shown in Figure 4C,F,I, we also observed that the frequencies are shifted from low to high values. This agrees with independent studies of noise effects in gene regulatory networks, where it has been shown that negative feedback loops shift noise frequency from low to high values compared to non-regulated circuits [56,57]. Therefore, we conclude that the shift to higher frequencies occurs due to a stronger influence of negative feedback loops in the larger systems that were analyzed in this work. This indicates that network wiring and architecture can influence the noise spectra. Remarkably, the noise suppression strategy in biological systems is different from standard frequency domain techniques that are commonly used in control systems engineering, electronics, and signal processing systems [58,59].

Interestingly, the spectral entropy value obtained for small nonlinear systems was smaller than spectral entropy values for the larger linear systems shown in Table 1. This may indicate that nonlinear systems can control and suppress noise more effectively than linear systems. However, the systems described by Equations (2) and (3) are very different and cannot be used for any conclusive comparison.

Power spectral density and spectral entropy are common tools that are often used to characterize electroencephalography and magnetic resonance encephalography recordings. For example, spectral analysis of electroencephalograms is used to study the neurophysiology of sleep [45], to detect differences in brain activities of subjects under normal and hypnosis conditions [46], and healthy subjects and schizophrenic patients [47]. Electroencephalography and magnetic resonance encephalography recordings in patients with drug-resistant epilepsy reveal the altered spectral entropy for electrophysiological and hemodynamic signals [48]. Spectral entropy for electroencephalographic signals can also be used to predict changes in memory performance [52]. We used the spectral analysis tools to characterize possible impacts of noise on signals generated by dynamical systems that are isomorphic with hypothetical specific conscious percepts.

Overall, our study showed that negative feedback loops in dynamical systems could suppress noise and shift it to a higher frequency. This property can be applied to neuronal dynamical systems that involve negative feedback loops; higher noise frequencies in a neuronal network can be more easily filtered out by other parts of the system that are composed of several connected subnetworks. We can also conclude that it is important to understand the contribution of noise to the dynamics of neural systems to successfully determine a minimal mechanism sufficient for any one specific conscious percept. Our study and analysis of a simple dynamical model that mimics a mechanism for neural correlates of consciousness revealed the particular impact of inherent fluctuations on the system and the influence of system size and architecture on noise spectra.

## Figures and Tables

**Figure 1 entropy-23-00583-f001:**
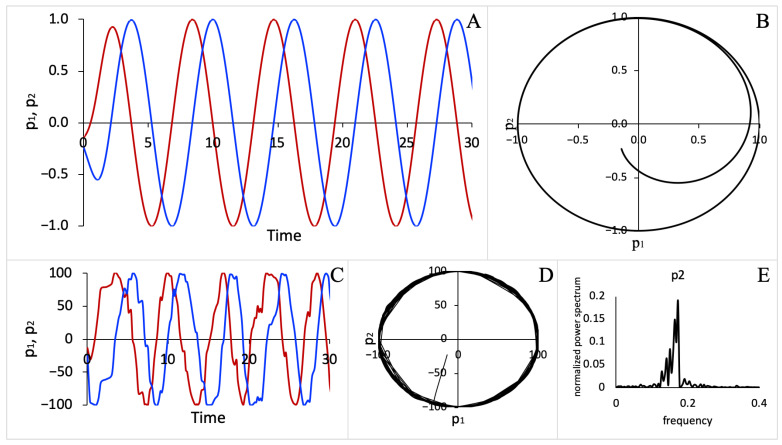
Dynamic behavior of a deterministic system (2) and the corresponding stochastic system. (**A**) Trajectories for processes *p*_1_ and *p*_2_ described by the system of differential equations (2), (**B**) the limit cycle in the *p*_1_*p*_2_-phase plane, (**C**) stochastic trajectories, (**D**) the limit cycle describing the solution of the stochastic system, (**E**) the normalized power spectral density characterizing the noise spectra in stochastic trajectories. The parameter *R* = 1 for the deterministic system (2) and *R* = 100 for the corresponding stochastic system.

**Figure 2 entropy-23-00583-f002:**
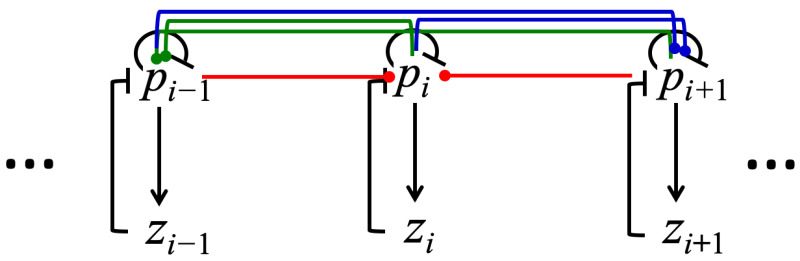
The influence diagram for processes described by the system of equations (3). Arrow-headed lines represent a positive influence and bar-headed lines represent a negative influence of one process on another or itself. The dot-headed lines represent positive or negative influence depending on the sign of the *ε* parameter. Different line colors are used for tracking purposes. Red lines represent interactions between $p_{i-1}$, $p_{i+1}$, and $p_{i}$; green lines represent interactions between $p_{i}$, $p_{i+1}$ and $p_{i-1}$; blue lines wire $p_{i-1}$, $p_{i}$ with $p_{i+1}$.

**Figure 3 entropy-23-00583-f003:**
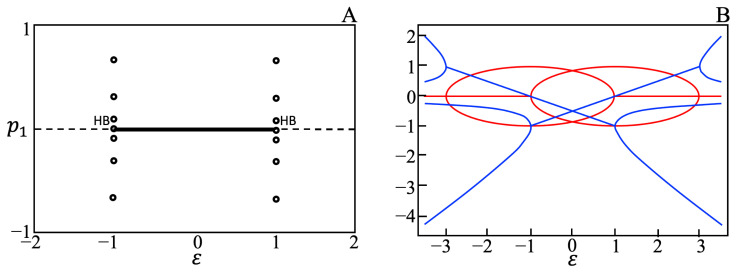
One parameter bifurcation diagram for the system of two $p_{i}$ and two $z_{i}$ processes (**A**), and real parts (blue curves) and imaginary parts (red curves) of eigenvalues as a function of parameter *ε* (**B**). Hopf bifurcation points (HB) were obtained at *ε* = ±1. The solid black line indicates the values of *ε* for which a spiral sink solution was obtained, the dashed black line indicates the values of *ε* for which a spiral source solution was observed, and open circles indicate periodic solutions. Further, the spiral sink solution was confirmed by the fact that real parts of all eigenvalues are negative for −1<ε<1 as shown in (**B**).

**Figure 4 entropy-23-00583-f004:**
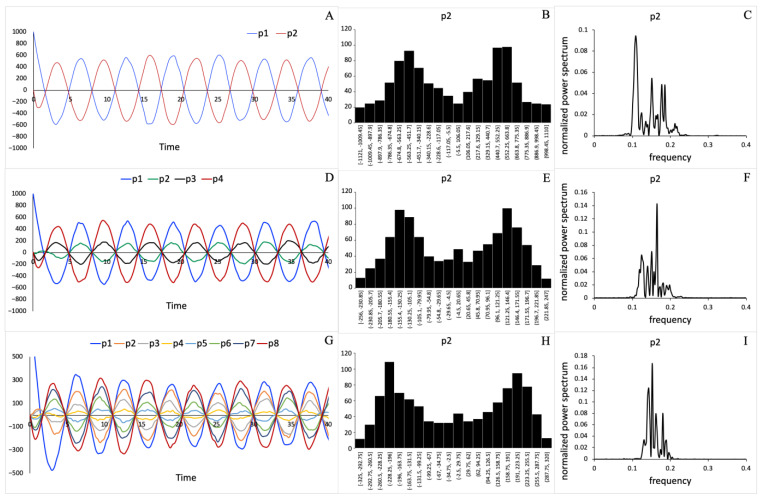
The implication of noise on system dynamics depends on system size (the number of processes in the system): (**A**,**D**,**G**) Stochastic trajectories for all processes $p_{i}$, (**B**,**E**,**H**) distribution histograms for process $p_{2},$ and (**C**,**F**,**I**) normalized power spectral densities for process $p_{2},$ which were obtained using systems of two, four, and eight $p_{i}$-processes, respectively. The power spectral density for a process $p_{i}$ depends on the number of processes constituting the system.

**Figure 5 entropy-23-00583-f005:**
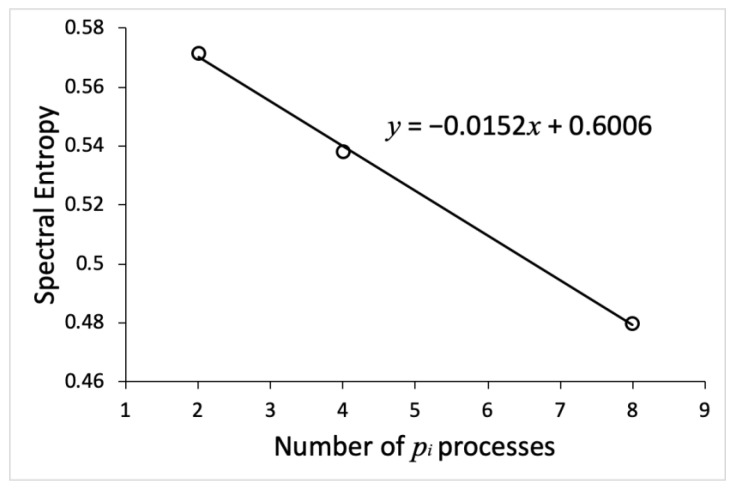
Spectral entropy decreases as a function of system size (the number of processes in the system). Open circles represent the average values of spectral entropy provided in Table 1. The solid line is a linear fit with the function displayed in the chart area.

**Table 1 entropy-23-00583-t001:** The dependence of spectral entropy for processes on system size.

**The System of Two *p_i_* Processes**		**The System of Eight *p_i_* Processes**	
**Process Name**	**Spectral Entropy**	**Process Name**	**Spectral Entropy**
*p* _1_	0.5735	*p* _1_	0.483
*p* _2_	0.5693	*p* _2_	0.466
The average entropy value = 0.5714		*p* _3_	0.474
**The System of Four *p_i_* Processes**		*p* _4_	0.512
**Process Name**	**Spectral Entropy**	*p* _5_	0.4986
*p* _1_	0.539	*p* _6_	0.4688
*p* _2_	0.542	*p* _7_	0.4686
*p* _3_	0.5375	*p* _8_	0.4664
*p* _4_	0.5343	The average entropy value = 0.48	
The average entropy value = 0.538			

## Data Availability

Data sharing not applicable.

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
