# Peer review of "Implications of Noise on Neural Correlates of Consciousness: A Computational Analysis of Stochastic Systems of Mutually Connected Processes"

_entropy, 2021, doi:10.3390/e23050583_

Round 1

Reviewer 1 Report

The paper is interesting. However, my main concern is that the paper may seem rather speculative. Hence I direct this suggestions and questions to the Author:

- From the abstract and the introduction I did not understand what is the narrative that links noise and consciousness. I think that the introduction is quite short and should deliver a stronger narrative that better highlights the thinking behind the paper. 
- Further, in the same line as above, the paper jumps to quickly to the results section. Indeed that happens in page 3. There should be a background section which expands on the literature review and explains used methods (such as spectral entropy and Guillespie's method).
- Some relevant research has been missed out. On the one hand there should be references linking the presented work to some actual application. Consider the case of human-operator modelling. In 'A new model of human steering using far-point error perception and multiplicative control' a stochastic model of human control is presented' and in 'Human response delay estimation and monitoring using gamma distribution analysis' stochastic response delays are considered. Also, I think the author will benefit from having a look at the literature on stochastic resonance, such as 'Stochastic resonance in noisy threshold neurons' and 'Adaptive stochastic resonance in noisy neurons based on mutual information'.
- One strong aspect of the paper is that the codes are provided to test this dynamic system in xppaut. 
- English grammar and style should be improved.

Author Response

The paper is interesting. However, my main concern is that the paper may seem rather speculative. Hence I direct this suggestions and questions to the Author:

We appreciate Reviewer’s constructive comments and suggestions. We have addressed all reviewer’s comments and carried out a comprehensive revision. We believe that our manuscript is significantly improved. Please find our answers to your comments below. Thank you.

- From the abstract and the introduction I did not understand what is the narrative that links noise and consciousness. I think that the introduction is quite short and should deliver a stronger narrative that better highlights the thinking behind the paper. 

We have expanded the introduction section to better link noise with consciousness. Reviewer’s suggestion to describe stochastic resonance in neuronal systems helped us to link noise with perception, consciousness and behavior.

We have described noise-induced transmission of information via stochastic resonance. We provided evidence that this has direct behavioral consequences and affects sensory information processing, perception and consciousness. We have also provided evidence that noise can increase information capacity of neuronal networks and maximize mutual information. We linked it with the information integration theory of consciousness, thus directly linking noise with conscious perception.

- Further, in the same line as above, the paper jumps to quickly to the results section. Indeed that happens in page 3. There should be a background section which expands on the literature review and explains used methods (such as spectral entropy and Guillespie's method).

We have added a background section on Guillespie's method and also a section that describes spectrum analysis and spectral entropy tools and their applications.

- Some relevant research has been missed out. On the one hand there should be references linking the presented work to some actual application. Consider the case of human-operator modelling. In 'A new model of human steering using far-point error perception and multiplicative control' a stochastic model of human control is presented' and in 'Human response delay estimation and monitoring using gamma distribution analysis' stochastic response delays are considered. Also, I think the author will benefit from having a look at the literature on stochastic resonance, such as 'Stochastic resonance in noisy threshold neurons' and 'Adaptive stochastic resonance in noisy neurons based on mutual information'.

We have added a section on stochastic resonance and also linked our work to some actual application.

- One strong aspect of the paper is that the codes are provided to test this dynamic system in xppaut. 

We appreciate this comment. Thank you.

- English grammar and style should be improved.

If our manuscript is accepted it will be sent to the writing center of Virginia Tech for the professional proofreading.

Reviewer 2 Report

The submitted article deals with understanding the implications of noise on neural correlates of consciousness through computational analysis of stochastic systems of mutually connected processes. The investigated topic is interesting and would definitely add value to existing knowledge base in this area. The article is well-structured and provides reasonably clear information about the fundamental theories of the proposed approach. However, I do feel that the following need tackling for the paper to better rounded in the discussed domain:

  1. there are several ongoing studies on the topic of signal noise suppression and handling, especially with rotating machine components where this is very common. Some authors have developed simplified ordinary spectrum as well as higher order spectrum based techniques, including the inclusion of coherence factors to suppress noise to great success. However, none of such articles or techniques were highlighted here. This in my opinion limits the inclusiveness of the review section. After all, those are very valued signal processing approaches, irrespective of areas of application. Please examine and consider the approaches described in references R1 and R2 as part of such techniques. R1 [An improved data fusion techniques for faults diagnosis  in  rotating machine] and R2 [integrated fault  detection  framework  for  classifying  rotating  machine  faults  using  frequency  domain  data  fusion  and  artificial  neural  networks.

  2. Can the authors describe the benefits/superiority of the proposed approach over standard frequency domain based approaches that have noise suppression abilities.

Author Response

The submitted article deals with understanding the implications of noise on neural correlates of consciousness through computational analysis of stochastic systems of mutually connected processes. The investigated topic is interesting and would definitely add value to existing knowledge base in this area. The article is well-structured and provides reasonably clear information about the fundamental theories of the proposed approach. However, I do feel that the following need tackling for the paper to better rounded in the discussed domain:

We thank Reviewer for constructive comments and suggestions. We have implemented all suggestions and believed that our manuscript is significantly improved. Please find our answers to your comments below. Thank you.

  1. there are several ongoing studies on the topic of signal noise suppression and handling, especially with rotating machine components where this is very common. Some authors have developed simplified ordinary spectrum as well as higher order spectrum based techniques, including the inclusion of coherence factors to suppress noise to great success. However, none of such articles or techniques were highlighted here. This in my opinion limits the inclusiveness of the review section. After all, those are very valued signal processing approaches, irrespective of areas of application. Please examine and consider the approaches described in references R1 and R2 as part of such techniques. R1 [An improved data fusion techniques for faults diagnosis  in  rotating machine] and R2 [integrated fault  detection  framework  for  classifying  rotating  machine  faults  using  frequency  domain  data  fusion  and  artificial  neural  networks.

 We have added a background section describing the application of spectrum-based techniques that are applied to quantify noise in biological and mechanical systems.

  1. Can the authors describe the benefits/superiority of the proposed approach over standard frequency domain based approaches that have noise suppression abilities

Biological circuits containing negative feedback loops are ubiquitous and present in gene regulatory circuits, molecular mechanisms and neuronal networks. We think that the suppression of noise by negative feedback loops may not be superior over standard frequency domain techniques used in signal processing systems, control systems engineering and electronics but could have evolutionary reasons (an analogous example that comes to mind is a fish using different swimming principles from a submarine, yet both are well able to move in the water). To highlight the difference between the standard frequency domain method and the biological approach, we have added a sentence in the discussion section.

Round 2

Reviewer 1 Report

The author has improved the manuscript, hence I recommend it for publication.